# Visual Imitation Learning with Recurrent Siamese Networks

## Abstract

People are incredibly skilled at imitating others by simply observing them. They achieve this even in the presence of significant morphological differences and capabilities. Further, people are able to do this from raw perceptions of the actions of others, without direct access to the abstracted demonstration actions and with only partial state information. People therefore solve a difficult problem of understanding the salient features of both observations of others and the relationship to their own state when learning to imitate specific tasks. We can attempt to reproduce a similar demonstration via trial and error and through this gain more understanding of the task space. To reproduce this ability an agent would need to both learn how to recognize the differences between itself and some demonstration and at the same time learn to minimize the distance between its own performance and that of the demonstration. In this paper we propose an approach using only visual information to learn a distance metric between agent behaviour and a given video demonstration. We train a Recurrent Neural Network (RNN)-based Siamese model to compute distances in space and time between motion clips while training an Rienforcement Learning (RL) policy to minimize this distance. Furthermore, we examine a particularly challenging form of this problem where the agent must learn an imitation based task given a single demonstration. We demonstrate our approach in the setting of deep learning based control for physical simulation of humanoid walking in both 2D with 10 degrees of freedom (DoF) and 3D with 38 DoF.

## 1 Introduction

Often in RL the designer formulates a reward function to elicit some desired behaviour in the policy. However, people often modify or refine their objectives as they learn. For example, a gymnast that is learning how to perform a flip can understand the overall motion from a few demonstrations. However, over time the gymnast, along with their previous experience, will learn to understand the less obvious but significant state features that determine a good flipping motion. In this same vein we want to gradually learn a distance function where, as the agent explores and gets more skilled, the agent refines its state space understanding and therefore the distance metric can further refine its accuracy.

Robots and people may plan using an internal pose space understanding; however, typically when people observe others performing tasks only visual information is available. Often, using distances in pose-space is ill-suited for imitation as changing some features will have result in drastically different visual appearance. In order to understand how to perform tasks from visual observation some mapping/transformation is used $pose = \phi(image)$, which allows for the minimization of $\phi(image) - agent_{pose}$. Even with a method to transform observations to a similar pose every person has different capabilities. Because of this, people must learn how to transform demonstrations into a representation where they can reproduce the behaviour to the best of their ability. In our work here we construct a distance metric derived from the agent's visual perceptions without the need for an intermediate pose representation by allowing the agent to observe itself externally and compare that perception with a demonstration.

Searching for a distance function has been an active topic of research (Abbeel & Ng, 2004; Argall et al., 2009). Given some vector of features the goal is to find an optimal transformation of these

features such that when differences are computed in this transformed space there exists a strong contextual meaning. For example, if we wanted a transformation that computed the distance between an agent's standing pose and its current pose, a good schema may prioritize the joint angles of the legs and torso while ignoring momentum. With a meaningful transformation function $\phi(\cdot)$ a distance can be computed between an agent and a demonstration. Previous work has explored the area of state-based distance functions, but many rely pose based metrics (Ho & Ermon, 2016; Merel et al., 2017a) that come from an expert. Few use image based inputs and none consider the importance of learning a distance function in time as well as space (Sermanet et al., 2017; Finn et al., 2017; Liu et al., 2017; Dwibedi et al., 2018). In this work we use a recurrent siamese network to learn the distance metric (Chopra et al., 2005).

An important detail of imitating demonstrations is their sequential and causal nature. There is both an ordering and speed in which the demonstration is performed. It is important to match the demonstrations state distribution. However, similarity between states may force the agent to imitate the same timing as the demonstration. This can be highly effective and lead to learning smooth motions but it also constrains the result to have similar timing as the demonstration. However, when the agent's motion becomes desynchronized with the demonstration the agent will receive low reward. Consider the case when a robot has learned to stand before it can walk. This pose exists inside the demonstration and should be encouraged. Therefore we learned an RNN-based distance function that can give reward for out of sync but similar behaviour. The work in Liu et al. (2017); Dwibedi et al. (2018); Sermanet et al. (2017) also performs imitation from video observation but each assumes some sort of time alignment between the agent and demonstration. Considering the data sparsity of the problem we include data from other tasks in order to learn a more robust distance function in visual sequence space.

Our method has similarities to both Inverse Reinforcement Learning (IRL) (Abbeel & Ng, 2004) and Generative Advisarial Imitation Learning (GAIL) (Ho & Ermon, 2016). The process of learning a cost function that will understand the space of policies in order to find an optimal policy given a demonstration is fundamentally IRL. While using positive examples from the expert and negative examples from the policy is similar to the method GAIL used to train a discriminator to understand the desired distribution. In this work we build upon these techniques by constructing a method that can learn polices using only noisy partially observable visual data. We also construct a cost function that takes into account the demonstration timing as well as pose by using a recurrent Siamese network. Our contribution rests on proposing and exploring this form of recurrent Siamese network as a way to address the key problem of defining the reward structure for imitation learning for deep RL agents.

## 2 PRELIMINARIES

In this section we outline some key details of the general RL framework and specific specialized formulations for RL that we rely upon when developing our method in Section: 3.

### 2.1 REINFORCEMENT LEARNING

Using the RL framework formulated with a Markov Dynamic Process (MDP): at every time step $t$, the world (including the agent) exists in a state $s_t \in S$, wherein the agent is able to perform actions $a_t \in A$, sampled from a policy $\pi(s_t, a_t)$ which results in a new state $s_{t+1} \in S$ according to the transition probability function $T(s_t, a_t, s_{t+1})$. Performing action $a_t$ from state $s_t$ produces a reward $r_t$ from the environment; the expected future discounted reward from executing a policy $\pi$ is:

$$J(\pi) = \mathbb{E}_{r_0,\dots,r_T}\left[\sum_{t=0}^{T} \gamma^t r_t\right] \tag{1}$$

where $T$ is the max time horizon, and $\gamma$ is the discount factor, indicating the planning horizon length.

The agent's goal is to optimize its policy, $\pi$, by maximizing $J(\pi)$. Given policy parameters $\theta_\pi$, the goal is reformulated to identify the optimal parameters $\theta_\pi^*$:

$$\theta_\pi^* = \arg\max_{\theta_\pi} J(\pi(\cdot|\theta_\pi)) \tag{2}$$

Using a Gaussian distribution to model the stochastic policy $\mu_{\theta_t}(s_t)$. Our stochastic policy is formulated as follows:

$$a_t \sim \pi(a_t \mid s_t, \theta_\pi) = \mathcal{N}(\mu(s_t \mid \theta_\pi), \Sigma) \qquad \Sigma = diag\{\sigma_i^2\} \tag{3}$$

where $\Sigma$ is a diagonal covariance matrix with entries $\sigma_i^2$ on the diagonal, similar to (Peng et al., 2017).

For policy optimization we employ stochastic policy gradient methods (Sutton et al., 2000). The gradient of the expected future discounted reward with respect to the policy parameters, $\nabla_{\theta_\pi} J(\pi(\cdot|\theta_\pi))$, is given by:

$$\nabla_{\theta_\pi} J(\pi(\cdot|\theta_\pi)) = \int_S d_\theta(s) \int_A \nabla_{\theta_\pi} \log(\pi(a, s|\theta_\pi)) A_\pi(s, a) \, da \, ds \tag{4}$$

where $d_\theta = \int_S \sum_{t=0}^T \gamma^t p_0(s_0)(s_0 \to s \mid t, \pi_0) \, ds_0$ is the discounted state distribution, $p_0(s)$ represents the initial state distribution, and $p_0(s_0)(s_0 \to s \mid t, \pi_0)$ models the likelihood of reaching state $s$ by starting at state $s_0$ and following the policy $\pi(a, s|\theta_\pi)$ for $T$ steps (Silver et al., 2014). $A_\pi(s, a)$ represents an advantage function (Schulman et al., 2016).

## 2.2 IMITATION LEARNING

Imitation learning is the process of training a new policy to reproduce the behaviour of some expert policy. Behavioural Cloning (BC) is a fundamental method for imitation learning. Given an expert policy $\pi_E$ possibly represented as a collection of trajectories $\tau < (s_0, a_0), \ldots, (s_T, a_T) >$ a new policy $\pi$ can be learned to match this trajectory using supervised learning.

$$\max_\theta \mathbb{E}_{\pi_E}[\sum_{t=0}^T log\pi(a_t|s_t, \theta_\pi)] \tag{5}$$

While this simple method can work well, it often suffers from distribution mismatch issues leading to compounding errors as the learned policy deviates from the expert's behaviour.

Similar to BC, IRL also learns to replicate some desired behaviour. However, IRL makes use of the environment, using the RL environment without a defined reward function. Here we describe maximal entropy IRL (Ziebart et al., 2008). Given an expert trajectory $\tau < (s_0, a_0), \ldots, (s_T, a_T) >$ a policy $\pi$ can be trained to produce similar trajectories by discovering a distance metric between the expert trajectory and trajectories produced by the policy $\pi$.

$$\max_{c \in C} \min_\pi (\mathbb{E}_\pi[c(s, a)] - H(\pi)) - \mathbb{E}_{\pi_E}[c(s, a)] \tag{6}$$

where $c$ is some learned cost function and $H(\pi)$ is a causal entropy term. $\pi_E$ is the expert policy that is represented by a collection of trajectories. IRL is searching for a cost function $c$ that is low for the expert $\pi_E$ and high for other policies. Then, a policy can be optimized by maximizing the reward function $r_t = -c(s_t, a_t)$.

GAIL (Ho & Ermon, 2016) uses a Generative Advasarial Network (GAN)-based (Goodfellow et al., 2014) framework where the discriminator is trained with positive examples from the expert trajectories and negative examples from the policy. The generator is a combination of the environment and the current state visitation probability induced by the policy $p_\pi(s)$.

$$\min_{\theta_\pi} \max_{\theta_\phi} \mathbb{E}_{\pi_E}[log(D(s, a|\theta_\phi))] + \mathbb{E}_{\pi_{\theta_\pi}}[log(1 - D(s, a|\theta_\phi))] \tag{7}$$

In this framework the discriminator provides rewards for the RL policy to optimize, as the probability of a state generated by the policy being in the distribution $r_t = D(s_t, a_t|\theta_\phi)$.

## 3 OUR APPROACH

In this section we describe our method to perform recurrent vision based imitation learning.

### 3.1 Partial Observable Imitation Learning Without Actions

For many problems we want to learn how to replicate the behaviour of some expert $\pi_E$ without access to the experts actions. Instead, we may only have access to an actionless noisy observation of the expert that we call a demonstration. Recent work uses BC to learn a new model to estimate the actions used via maximum-likelihood estimation (Torabi et al., 2018). Still, BC often needs many expert examples and tends to suffer from state distribution mismatch issues between the *expert* policy and the *student* (Ross et al., 2011). Work in (Merel et al., 2017b) proposes a system based on GAIL that can learn a policy from a partial observation of the demonstration. In this work the state input to the discriminator is a customized version of the expert's pose and does not take into account the demonstration's sequential nature. The work in (Wang et al., 2017) provides a more robust GAIL framework along with a new model to encode motions for few-shot imitation. This model uses an RNN to encode a demonstration but uses expert state and action observations. In our work we limit the agent to only a partial visual observation as a demonstration. Additional works learn implicit models of distance (Yu et al., 2018; Pathak et al., 2018; Finn et al., 2017; Sermanet et al., 2017), none of these explicitly learn a sequential model to use the demonstration timing. Another version of GAIL, infoGAIL (Li et al., 2017), was used on some pixel based inputs. In contrast, here we train a recurrent siamese model that can be used to enable curriculum learning and allow for reasonable distances to be computed even when the agent and demonstration are out of sync or have different capabilities.

### 3.2 Distance-Based Reinforcement Learning

Given a distance function $d(s)$ that indicates how far away the agent is from some desired behaviour a reward function over states can be constructed $r(s) = -d(s)$. In this framework there is no reward signal coming from the environment instead fixed rewards produced by the environment are replaced by the agent's learned model that is used to compare itself to some desired behaviour.

$$J(\pi) = \mathbb{E}_{d(s_0),...,d(s_T)} \left[ \sum_{t=0}^{T} \gamma^t (-d(s_t)) \right] \tag{8}$$

Here $d(s)$ can take many forms. In IRL it is some learned cost function, in GAIL it is the discriminator probability. In this framework this function can be considered more general and it can be interpreted as *distance from desired behaviour*, Specifically, in our work $d(s)$ is learning a distance between video clips.

Many different methods can be used to learn a distance function in state-space. We use a standard triplet loss over time and task data Chopra et al. (2005). The triplet loss is used to minimize the distance between two examples that are *positive*, very similar or from the same class, and maximize the distance between pairs of examples that are known to be un-related.

Data used to train the siamese network is a combination of trajectories $\tau = \langle s_0, \ldots, s_T \rangle$ generated from simulating the agent in the environment as well as the demonstration.

$$\mathcal{L}(s_i, s_p, s_n) = y * ||f(s_i) - f(s_p)|| + ((1 - y) * (\max(\rho - (||f(s_i) - f(s_n)||), 0))) \tag{9}$$

Where $y = 1$ is a positive example $s_p$, pair where the distance should be minimal and $y = 0$ is a negative example $s_n$, pair where the distance should be maximal. The *margin* $\rho$ is used as an attractor or anchor to pull the negative example output away from $s_i$ and push values towards a 0 to 1 range. $f(\cdot)$ computes the output from the underlying network. A diagram of this image-based training process and design is shown in Figure 1a. The distance between two states is calculated as $d(s, s') = ||f(s) - f(s')||$ and the reward as $r(s, s') = -d(s, s')$. For recurrent models we use the same loss however, the states $s_p, s_n, s_i$ are sequences. The sequence is fed into the RNN and a single output encoding is produced for that sequence. During RL training we compute a distance given the sequence of states observed so far in the episode. This is a very flexible framework that allows us to train a distance function in state space where all we need to provide is labels that denote if two states, or sequences, are similar or not.

### 3.3 Sequence Imitation

Using a distance function in the space of states can allow us to find advantageous policies. The hazard with using a state only distance metric when you are given demonstrations as sequences to

imitate is that the RL agent can suffer from phase-mismatch. In this situation the agent may be performing the desired behaviour but at a different speed. As the demonstration timing and agent diverge the agent receives less reward, even though it is visiting states that exist elsewhere in the demonstration. If instead we consider the current state conditioned on the previous states, we can learn to give reward for visiting states that are only out of sync with the demonstration motion.

This distance metric is formulated in a recurrent style where the distance is computed from the current state and conditioned on all previous states $d(s_t|s_{t-1}, \ldots, s_0)$. The loss function remains the same as in Eq. 9 but the overall learning process changes to use an RNN-based model. A diagram of the method is shown in Figure 1b. This model uses a time distributed RNN. A single convolutional network $conv^a$ is first used to transform images of the agent and demonstration to encoding vectors $e^a t$. After the sequence of images is distributed through $conv^a$ there is an encoded sequence $< e_0^a, \ldots, e_t^a >$, this sequence is fed into the RNN $lstm^a$ until a final encoding is produced $h_t^a$. This same process is done for a copy of the RNN $lstm_b$ producing $h_t^b$ for the demonstration. The loss is computed in a similar fashion to (Mueller & Thyagarajan, 2016) using the sequence outputs of images from the agent and another from the demonstration. The reward at each timestep is computed as $r_t = ||h_t^a - h_t^b||$. At the beginning of ever episode the RNN's internal state is reset. The policy and value function use a 2 layer neural network with 512 and 256 units respectively.

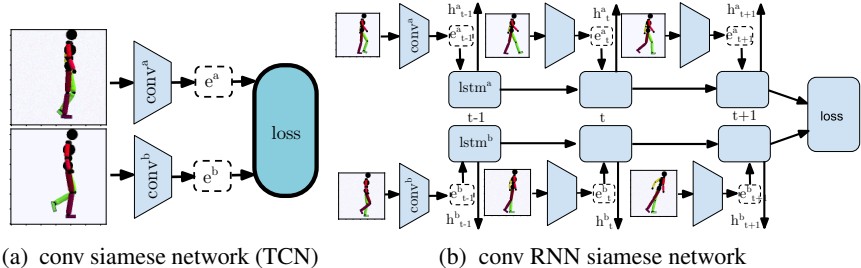

(a) conv siamese network (TCN)  (b) conv RNN siamese network

Figure 1: Siamese network network structure. The convolutional portion of the network includes 3 convolution layers of 16 filters with size $10 \times 10$ and stride $5 \times 5$, 32 filters of size $5 \times 5$ and stride $2 \times 2$ and 32 filters of size $3 \times 3$ and stride $1 \times 1$. The features are then flattened and followed by two dense layers of 256 and 128 units. The majority of the network uses ReLU activations except the last layer that uses a sigmoid activation. Dropout is used between the convolutional layers. The RNN-based model uses a GRU layer with 128 hidden units, followed by a dense layer of 128 units.

## 3.4 THE RL SIMULATION ENVIRONMENT

Our simulation environment is similar to the OpenAIGym robotics environments (Plappert et al., 2018). The demonstration $M$ the agent is learning to imitate is generated from a clip of mocap data. The mocap data is used to animate a second robot in the simulation. Frames from the simulation are captured and used as video input to train the distance metric. The policy is trained on pose data, as link distances and velocities relative to the robot's Centre of Mass (COM) of the simulated robot. This is a new simulation environment that has been created to take motion capture data and produce multi-view video data that can be used for training RL agents or generating data for computer vision tasks. The environment also includes new challenging and dynamic tasks for humanoid robots. The simulation and rendering have been optimized and efficient EGL-based off-screen rendering is used to capture video data.

## 3.5 DATA AUGMENTATION

In a manner similar to how a person may learn to understand and reproduce a behaviour (Council et al., 2000; Gentner & Stevens, 2014) we apply a number of data augmentation methods to produce additional data used to train the distance metric. Using methods analogous to the cropping and warping methods popular in computer vision (He et al., 2015) we randomly *crop* sequences and randomly *warp* the demonstration timing. The *cropping* is performed by both initializing the agent to random poses from the demonstration motion and terminating episodes when the agent's head, hands or torso contact the ground. The motion *warping* is done by replaying the demonstration

motion at different speeds. We also make use of time information in a similar way to (Sermanet et al., 2017), where observations at similar times in the same sequence are often correlated and observations at different times may have little similarity. To this end we generate more training samples using randomly cropped sequences from the same trajectory as well as reversed and out of sync versions. Imitation data for other tasks is also used to help condition the distance metric learning process. Motion clips for running, backflips and frontflips are used along with the desired walking motion. See the Appendix for more details on how positive and negative pairs are created from this data.

The algorithm used to train the distance metric and policy is outlined in Algorithm 1. Importantly the RL environment generates two different state representations for the agent. The first state $s_{t+1}$ is the internal robot pose. The second state $s_{t+1}^v$ is the agent's rendered view. The rendered view is used with the distance metric to compute the similarity between the agent and the demonstration. We attempted to use the visual features as the state input for the policy as well. This resulted in poor policy quality.

---

**Algorithm 1** Visual Imitation Learning Algorithm

---

1:  Randomly initialize model parameters $\theta_\pi$ and $\theta_d$
2:  Create experience memory $D \leftarrow \{\}$
3:  Given a demonstration $M \leftarrow <m_0, \ldots, m_T>$
4:  **while** not done **do**
5:      **for** $i \in \{0, \ldots N\}$ **do**
6:          $\tau_i \leftarrow \{\}$
7:          **for** $t \in \{0, \ldots, T\}$ **do**
8:              $a_t \leftarrow \pi(\cdot|s_t, \theta_\pi)$
9:              $s_{t+1}, s_{t+1}^v \leftarrow env(a_t)$
10:             $r_t \leftarrow -d(s_{t+1}^v, m_{t+1}|\theta_d)$
11:             $\tau_{i,t} \leftarrow <s_t, a_t, r_t>$
12:             $s_t \leftarrow s_{t+1}$
13:         **end for**
14:     **end for**
15:     $D \leftarrow D \bigcup \{\tau_0, \ldots, \tau_N\}$
16:     Update the distance metric $d(\cdot)$ parameters $\theta_d$ using $D$
17:     Update the policy parameters $\theta_\pi$ using $\{\tau_0, \ldots, \tau_N\}$
18: **end while**

---

# 4 Experiments and Results

This section contains a collection of experiments and results produced to investigate the capabilities of the method.

## 4.1 2D Humanoid Imitation

The first experiment performed uses the method to learn a cyclic walking gait for a simulated humanoid walking robot given a single motion example, similar to (Peng & van de Panne, 2017). In this simulated robotics environment the agent is learning to imitate a given reference motion of a walk. The agent's goal is to learn how to actuate Proportional Derivative (PD) controllers at 30 fps to mimic the desired motion. The simulation environment provides a hard coded reward function based on the robot's pose that is used to evaluate the policies quality. The images captured from the simulation are converted to grey-scale with $128 \times 128$ pixels. In between control timesteps the simulation collects images of the agent and the rendered demonstration motion. The agent is able to learn a robust walking gate even though it is only given noisy partial observations of a demonstration. We find it extremely helpful to *normalize* the distance metric outputs using $r = exp(r^2 * w_d)$ where $w_d = -5.0$ scales the filtering width (Peng & van de Panne, 2017). Early in training the distance metric often produces noisy large values, also the RL method constantly updates reward scaling statistics, the initial high variance data reduces the significance of better distance metric values produced later on by scaling them to very small numbers. The improvement of using this normalize

reward is shown in Figure 3a. Example motion of the agent after learning is shown in Figure 2 and in the supplemental Video.

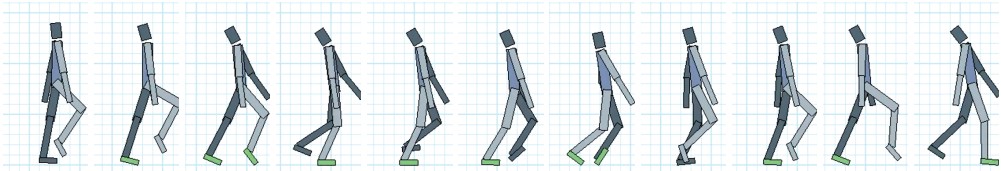

Figure 2: Still frame shots from trained policy in the humanoid2d environment.

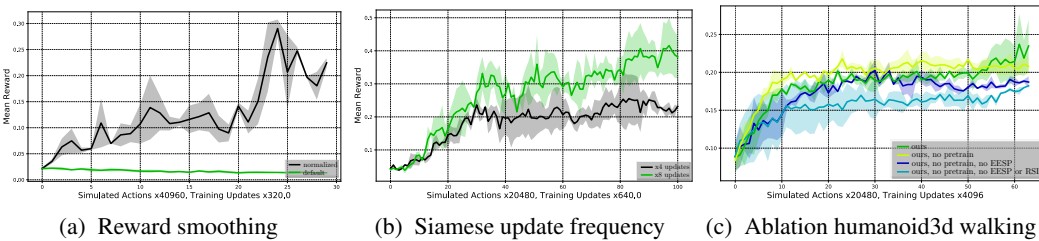

(a) Reward smoothing     (b) Siamese update frequency     (c) Ablation humanoid3d walking

Figure 3: Ablation analysis of the method. We find that training RL policies is sensitive to size and distribution of rewards. The siamese network benefits from a number of training adjustments that make it more suitable for RL.

## 4.2 ALGORITHM ANALYSIS

We compare the method to two other methods that can be considered as learning a distance function in state space, GAIL and using a Variational Auto Encoder (VAE) to train an encoding and compute distances between those encodings using the same method as the siamese network in Figure 4a. We find that the VAE method does not appear to capture the important distances between states, possibly due to the complexity of the decoding transformation. Similarly, we use try a GAIL type baseline and find that the method would either produce very jerky motion or stand still, both of which are contained in the imitation data. Our method that considers the temporal structure of the data produces higher value policies.

Additionally, we create a multi-modal version of the method where the same learning framework is use. Here we replace the bottom conv net with a dense network and learn a distance metric between agent poses and imitation video. The results of these models along with using the default manual reward function are shown in Figure 4b. The multi-modal version appears to perform about equal to the vision-only modal. In Figure 4b and Figure 8c we compare our method to a non sequence based model that is equivalent to TCN. On average the method achieves higher value policies.

We conduct an additional ablation analysis in Figure 3c to compare the effects of particular methods used to assist in training the recurrent siamese network We find it very helpful to reduce Reference State Initialization (RSI). If more episodes start in the same state it increases the temporal alignment of training batches for the RNN. We believe it is very important that the distance metric be most accurate for the earlier states in an episode so we use Early Episode Sequence Priority (EESP). Meaning we give the chances of cropping the window used for RNN training batches closer to the beginning of the episode. Also, we give higher probability to shorter windows. As the agent gets better the average length of episodes increases and so to will the average size of the cropped window. Last, we tried pretraining the distance function. This leads to mixed results, see Figure 7a and Figure 7b. Often, pretraining overfits the initial data collected leading to poor early RL training. However, in the long run pretraining does appear to improve over its non pretrained version.

## 4.3 3D HUMANOID ROBOT IMITATION

In these simulated robotics environments the agent is learning to imitate a given reference motion of a walk, run, frontflip or backflip. A single imitation motion demonstration is provided by the simula-

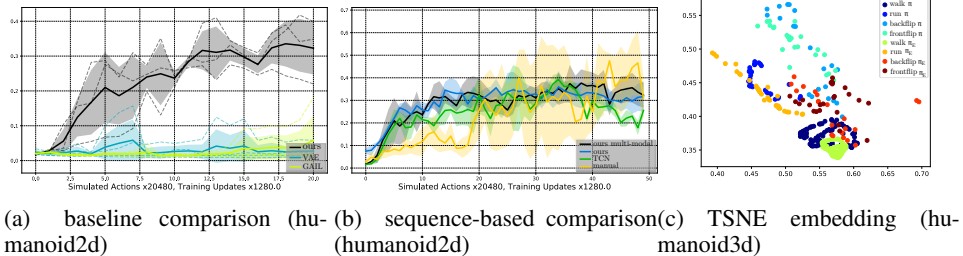

(a) baseline comparison (humanoid2d)  (b) sequence-based comparison (humanoid2d)  (c) TSNE embedding (humanoid3d)

Figure 4: Baseline comparisons between our sequence-based method, GAIL and TCN. In 4a We compare our method to GAIL and a VAE where use using the euclidean distance of the encodings. We perform two additional baseline comparison between out method and TCN in 4b and 8c. These both show that on average our method performs similar to TCN or better over time. In these plots the large solid lines are the average performance of a collection of policy training run. The dotted lines of the same colour are the specific performance value for each policy training run. The filled in areas around the average policy performance is the variance other the collection of policy training runs.

tion environment as a cyclic motion, similar to (Peng et al., 2018). The agent controls and observes frames at 30 fps. During learning, additional data is included from other tasks for the walking task this would be: walking-dynamic-speed, running, frontflips and backflips) that are used to train the distance metric. We also include data from a modified version of the tasks that has a randomly generated speed modifier $\omega \in [0.5, 2.0]$ walking-dynamic-speed, that warps the demonstration timing. This additional data is used to provide a richer understanding of distances in space and time. The input to the distance metric is a pair of frames from the simulation per control step, see Algorithm 1.

We find that using the RNN-based distance metric makes the learning process more gradual and smoother. This can be seen in Figure 4b where the original manually created reward can still provide sparse feedback after the agent has sufficiently diverged from the desired behaviour. Some example trajectories using the policy learned using the method are shown in Figure 5, Figure 6 and in the supplemental Video. We include a rendered version of the running task in Figure 6.

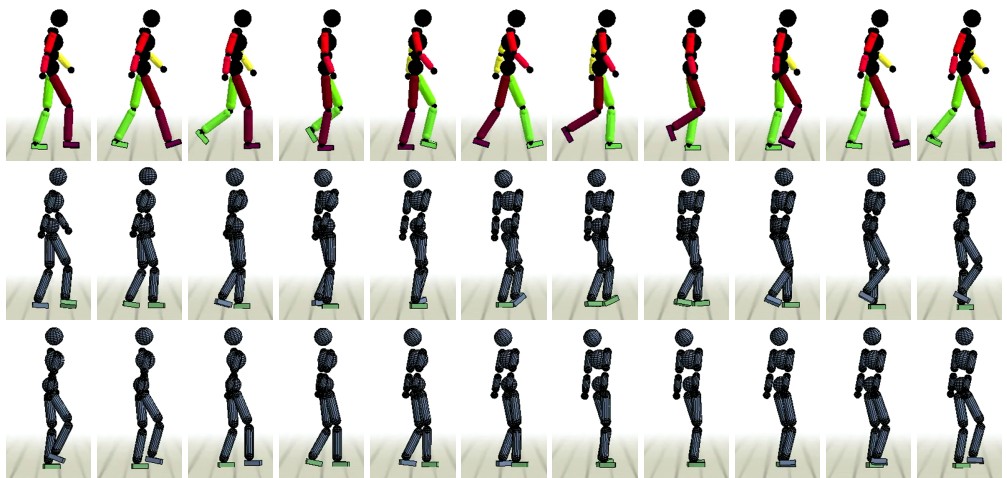

Figure 5: Still frame shots of the agent's motion after training on humanoid3d walking.

**Sequence Encoding** Using the learned sequence encoder a collection of motions from different classes are processed to create a TSNE embedding of the encodings (Maaten & Hinton, 2008). In Figure 4c we plot motions both generated from the learned policy $\pi$ and the expert trajectories $\pi_E$. Interestingly, there are clear overlaps in specific areas of the space for similar classes across learned $\pi$ and expert $\pi_E$ data. There is also a separation between motion classes in the data.

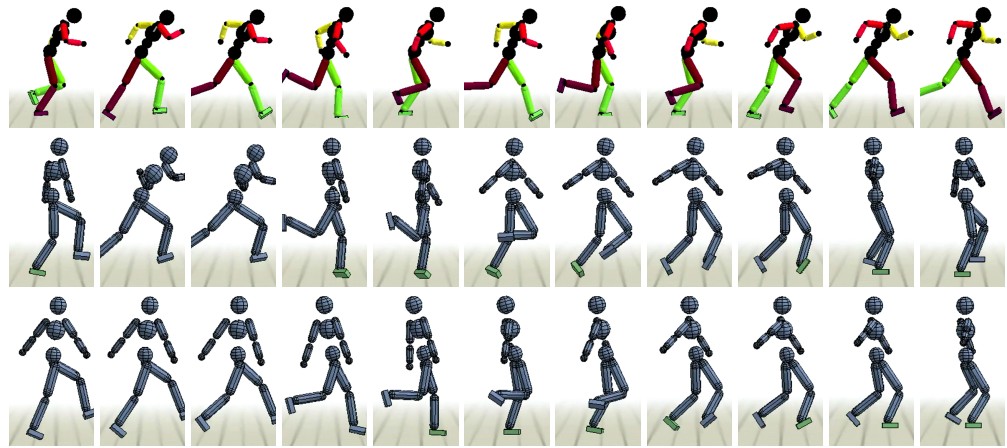

Figure 6: Still frame shots of the agent's motion after training on humanoid3d running.

## 5 DISCUSSION AND CONCLUSION

Learning a distance metric is imperfect, the distance metric can compute inaccurate distances in areas of the state space it has not yet been trained on. This could imply that when the agent explores and finds truly *new* and promising trajectories the distance metric will produce bad results. We attempt to mitigate this affect by including training data from different tasks while training the distance metric. We believe the method will benefit greatly from a larger collection of multitask data and increased variation of each task. Additionally, if the distance metric confidence is modelled this information could be used to reduce variance and overconfidence during policy optimization.

It appears Deep Deterministic Policy Gradient (DDPG) works well for this type of problem. Our hypothesis is the learned reward function is changing between data collection phases it may be better to view this as off-policy data. Learning a reward function while training also adds additional variance to the policy gradient. This may indicate that the bias of off-policy methods could be preferred over the added variance of on-policy methods. We also find it important to have a small learning rate for the distance metric Figure 7c. This reduces the reward variance between data collection phases and allows learning a more accurate value function. Another approach may be to use partially observable RL that has the ability to learn a better model of the value function given a changing RNN-based reward function. Training the distance metric could benefit from additional regularization such as constraining the kl-divergence between updates to reduce variance. Another option is learn a sequence-based policy as well given that the rewards are now not dependant on a single state observation.

We tried using GAIL but we found it has limited temporal consistency. This led to learning very jerky and overactive policies. The use of a recurrent discriminator for GAIL may mitigate some of these issues. It is challenging to produce result better than the carefully manually crafted reward functions used by some of the RL simulation environments that include motion phase information in the observations (Peng et al., 2018; 2017). Still as environments become increasing more realistic and grow in complexity we will need more complex reward functions to describe the desired behaviour we want from the agent.

Training the distance metric is a complex balancing game. One might expect that the model should be trained early and fast so that it quickly understands the difference between a good and bad demonstration. However, quickly learning confuses the agent, rewards can change quickly which can cause the agent to diverge off toward an unrecoverable policy space. Slower is been better, as the distance metric my not be accurate but it may be locally or relatively reasonable which is enough to learn a good policy. As learning continues these two optimizations can converge together.

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

# 6 APPENDIX

## 6.1 DATASETS

The mocap used in the created environment come from the CMU mocap database and the SFU mocap database.

## 6.2 TRAINING DETAILS

The learning simulations are trained using Graphics Processing Unit (GPU)s. The simulation is not only simulating the interaction physics of the world but also rendering the simulation scene in order to capture video observations. On average it takes 3 days to execute a single training simulation. The process of rendering and copying the images from the GPU is one of the most expensive operations with the method.

## 6.3 DISTANCE FUNCTION TRAINING

In Figure 7b we show the training curve for the recurrent siamese network. The model learns smoothly considering that the training data used is constantly changing as the RL agent explores. In Figure 7a the learning curve for the siamese RNN is shown after performing pretraining. We can see the overfitting portion the occurs during RL training. This overfitting can lead to poor reward prediction during the early phase of training.

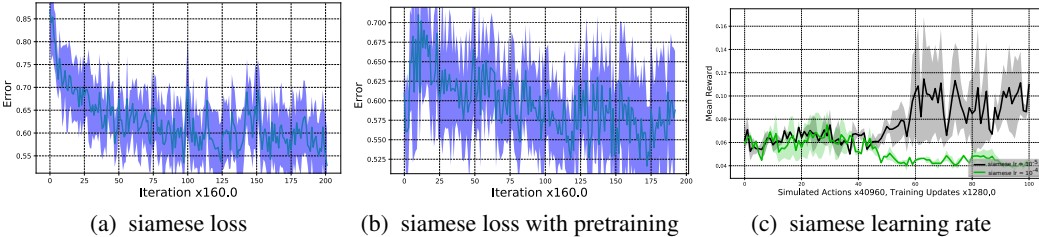

(a) siamese loss     (b) siamese loss with pretraining     (c) siamese learning rate

Figure 7: Training losses for the siamese distance metric. Higher is better as it indicates the distance between sequences from the same class are closer.

It can be difficult to train a sequenced based distance function. One particular challenge is training the distance function to be accurate across the space of possible states. We found a good strategy was to focus on the beginning of episode data. If the model is not accurate on states seen earlier in the episode it may never learn how to get into good states later on in the episode that the distance function understands better. Therefore, when constructing batches to train the RNN on we give higher probability to starting earlier in episodes. We also give a higher probability to shorter sequences. As the agent gets better average episodes length increase, so to will the randomly selected sequence windows.

## 7 Positive and Negative Examples

We use two methods to generate positive and negative examples. The first method is similar to TCN where we can make an assumption that sequences that overlap more in time are more similar. For each episode two sequences are generate, one for the agent and one for the imitation motion. We compute positive pairs by altering one of these seqeunces and comparing this altered verion to its original version. Here we list the number of ways we alter sequences for positive pairs.

1. Adding Gaussian noise to each state in the sequence (mean $= 0$ and variance $= 0.02$)
2. Out of sync versions where the first state is removed from the first sequence and the last state from the second sequence
3. Duplicating the first state in either sequence
4. Duplicating the last state in either sequence

We alter sequences for negative pairs by

1. Reversing the ordering of the second sequence in the pair.
2. Randomly picking a state out of the second sequence and replicating it to be as long as the first sequence.
3. Randomly shuffling one sequence.
4. Randomly shuffling both sequences.

The second method we use to create positive and negative examples is by including data for additional classes of motion. These classes denote different task types. For the humanoid3d environment we generate data for walking-dynamic-speed, running, backflipping and frontflipping. Pairs from the same tasks are labelled as positive and pair from different classes are negative.

### 7.1 RL Algorithm Analysis

It is not clear which RL algorithm may work best for this type of imitation problem. A number of RL algorithms were evaluated on the humanoid2d environment Figure 8a. Surprisingly, Trust Region Policy Optimization (TRPO) (Schulman et al., 2015) did not work well in this framework, considering it has a controlled policy gradient step, we thought it would reduce the overall variance. We found that DDPG (Lillicrap et al., 2015) worked rather well. This could be related to having a changing reward function, in that if the changing rewards are considered off-policy data it can be easier to learn. This can be seen in Figure 8b where DDPG is best at estimating the future discounted rewards in the environment. We tried also Continuous Actor Critic Learning Automaton (CACLA) (Van Hasselt, 2012) and Proximal Policy Optimization (PPO) (Schulman et al., 2017), we found that PPO did not work particularly well on this task, this could also be related to added variance.

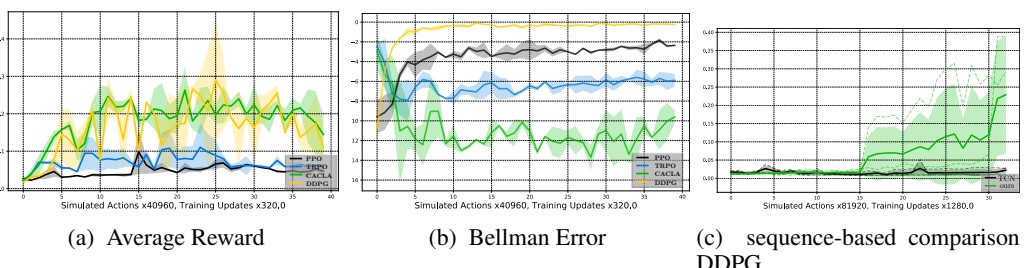

(a)  Average Reward

(b)  Bellman Error

(c)  sequence-based  comparison DDPG

Figure 8: RL algorithm comparison on humanoid2d environment.

