# OpenReview forum: "Visual Imitation Learning with Recurrent Siamese Networks"
_ICLR.cc/2019/Conference_

### Official Review · AnonReviewer1 · 2018-10-31
**Interesting idea, writing needs significant work and comparisons**

**Rating:** 5
**Confidence:** 4

**Review:**


Brief summary: This work proposes a way to perform imitation learning from raw videos of behaviors, without the need for any special time-alignment or actions present. They are able to do this by using a recurrent siamese network architecture to learn a distance function, which can be used to provide rewards for learning behaviors, without the need for any explicit pose estimation. They demonstrate effectiveness on 2 different locomotion domains.

Overall impression:
Overall, my impression from this paper is that the idea is to use a recurrent siamese network to learn distances which make sense in latent space and provide rewards for RL. This is able to learn interesting behaviors for 2 tasks. But I think the writing needs significant work for clarity and completeness, and there needs to be many more baseline comparisons.

Abstract comments:
trail and error -> trial and error

Introduction comments:

Alternative reasons why pose estimation won’t work is because for any manipulation tasks, you can’t just detect pose of the agent, you also have to detect pose of the objects which may be novel/different

Few use image based inputs and none consider the importance of learning a distance function in time as well as space -> missed a few citations (eg imitation from observation (Liu, Gupta, et al))

Therefore we learned an RNN-based distance function that can give reward for out of sync but similar behaviour -> could be good to emphasize difference from imitation from observation (Liu, Gupta, et al) and TCN (Semanet et al), since they both assume some sort of time alignment

Missing related work section. There is a lot of related work at this point and it is crucial to add this in. Some things that come to mind beyond those already covered are:
1. Model-based Imitation Learning from State Trajectories
2. Reward Estimation via State Prediction
3. infoGAIL
4. Imitation from observation
5. SFV: Reinforcement Learning of Physical Skills from Videos
6. Universal planning networks
7. https://arxiv.org/abs/1808.00928
8. This might also be related to VICE (Fu, Singh et al), in that they also hope to learn distances but for goal images only.
It seems like there is some discussion of this in Section 3.1, but it should be it’s own separate section.

Section 3 comments:
a new model can be learned to match this trajectory using some distance metric between the expert trajectories and trajectories produced by the policy π -> what does this mean. Can this be clarified?
 The first part of Section 3 belongs in preliminaries. It is not a part of the approach.

Section 3.2
Equations 9 and 10 are a bit unnecessary, take away from the main point

What does distance from desired behaviour mean? This is not common terminology and should be clarified explicitly.

Equation 11 is very confusing. The loss function is double defined.  what exactly Is the margin \rho (is it learned?) The exact rationale behind this objective, the relationship to standard siamese networks/triplet losses like TCN should be discussed carefully. This is potentially the most important part of the paper, it should be discussed in detail.Also is there a typo, should it be || f(si) - f(sn)|| if we want it to be distances? Also the role of trajectories is completely not discussed in equation 11.

Section 3.3
The recurrent siamese architecture makes sense, but what the positive and negative examples are, what exactly the loss function is, needs to be defined clearly. Also if there are multiple demonstrations of a task, which distance do we use then?

The RL simulation environment is it made in-house, based on bullet or something else?

Data augmentation - how necessary is this for method success? Can an ablation be done to show the necessity of this?

Algorithm 1 has some typos
- > is missing in line 3
- Describe where reward r is coming from in line 10

Section 4.1
Walking gate -> walking gait

There are no comparisons with any of the prior methods for performing this kind of thing. For example, using the pose estimation baseline etc. Using the non-recurrent version. Using TCN type of things. It’s not hard to run these and might help a lot, because right now there are no baseline comparisons

---

> ### Author Response · Authors · 2018-11-21
> **Specific Comments**
>
> re: meaning of "desired behaviour"
>
> We are trying to get away from saying the video, movie or motion. We are really trying to learn something more abstract here. We are using videos in this case, we will be more clear in the paper but we are trying to highlight the temporal complexity of imitation.
>
> re: The RL simulation environment is it made in-house, based on bullet or something else?
>
> It is based on Bullet and is made in-house and will be released with this work.

---

### Official Review · AnonReviewer2 · 2018-11-02
**An interesting solution to a challenging problem, but lacking in quantitative results**

**Rating:** 4
**Confidence:** 3

**Review:**

Summary: This paper aims to imitate, via Imitation Learning, the actions of a humanoid agent given only video demonstrations of the desired task, including walking, running, back-flipping, and front-flipping. Since the algorithm does not have direct access to the underlying actions or rewards, the agent aims to learn an embedding space over instances with the hope that distance in this embedding space corresponds to reward. Deep RL is used to optimize a policy to maximize cumulative reward in an effort to reproduce the behavior of the expert.

High-level comments:
- My biggest concern with this paper is the lack of a baseline example that we can use to evaluate performance. The walking task is interesting, but the lack of a means by which we can evaluate a comparison between different approaches makes it very difficult to optimize. This makes evaluation of quality and significance rather difficult. A number of other questions I have stem from this concern:
    = The paper is missing a comparison between the recurrent Siamese network and the non-recurrent Siamese network. The difficulty in comparing these approaches without a quantitative performance metric.
    = The authors also mention that they tried using GAIL to solve this problem, but do not show these results. Again, a success metric would be very helpful here.
    = Finally, a simpler task for which the reward is more easily specified may be a better test case for the quantitative results. Right now, the provided example of walking agents seems to only provide quantitative results.
- The authors need to be more clear about the structure of the training data and the procedure. As written, the structure of the triplet loss is particular ambiguous: the condition for positive/negative examples is not clearly specified.
- There are a number of decisions made in the paper that feel rather arbitrary or lack justification. In particular, the "normalization" scaling factor fits into this category. Some intuition or explanation for why this is necessary (or why this functional form should be preferred) would be helpful.
- A description of what the error bars represent in all of the plots is necessary.

More minor comments and questions:
- The choice of RL algorithm is not the purpose of this paper. Much of this section, and perhaps many of the training curves, are probably better suited to appear in the Appendix. Relatedly, why are training curves only shown for the 2D environment? If space was a concern, the appendix should probably contain these results.
- An additional agent that may be a useful comparison is one that is directly provided the actions. It might then be more clear how well. (Again, this would require a way to compare performance between different approaches.)
- How many demonstrations are there? At training vs testing?
- Where are the other demonstrations? The TSNE embedding plot mentions other tasks which do not appear in the rest of the paper. Did these demonstrations not work very well?

A Comment on Quality: Right now, the paper needs a fair bit of cleaning up. For instance, the word "Rienforcement" is misspelled in the abstract. There is also at least one hanging reference. Finally, a number of references need to be added. For example, when the authors introduce GAIL, they mention GANs and cite Goodfellow et al. 2014, but do not cite GAIL. There is also a lot of good research on Behavioral Cloning, and where it can go wrong, that the authors mention, but do not cite.

Conclusion: At this point it is difficult to recommend this paper for acceptance, because it is very hard to evaluate performance of the technique. With a more concrete way of evaluating performance on a different task with a clearer reward function for comparison, the paper could be much stronger, because this would allow the authors to compare the techniques they propose to one another and to other algorithms (like GAIL).

---

> ### Author Response · Authors · 2018-11-21
> **Sqecific comments**
>
> re: reward "normalization"
>
> Good question. I believe this has to do with the initial outputs from the siamese network. When training starts the values can be between -50 and 50 which is rather large. The RL method constantly updates reward normalization statistics so the initial high variance data reduces the significance of better distance metric values produced later on by scaling them to very small numbers...
>
> re: training details.
> We are editing these now.

---

### Official Review · AnonReviewer3 · 2018-11-07
**Issues with significance of results**

**Rating:** 4
**Confidence:** 4

**Review:**

This paper proposes an imitation learning method solely from video demonstrations by learning recurrent image-based distance model and in conjunction using RL to track that distance.

Clarity: The paper writing is mostly clear. The motivation for using videos as a demonstration source could be more clearly stated. One reason is because it would pave the way to learn from real-world video demonstrations. Another reason is that robot's state space is an ill-suited space to be comparing distances over and image space is more suitable. Choosing one would help the readers identify the paper's motivation and contribution.

Originality: The individual parts of this work (siamese networks, inverse RL, learning distance functions for IRL, tracking from video) have all been previously studied (which would be good to discuss in a relate work section), so nothing stands out as original, however the combination of existing ideas is well-chosen and sensible.

Significance: There are a number of factors that limit the significance of this work.

First, the demonstration videos come from synthetic rendered systems very similar the characters that imitate them, making it hard to evaluate whether this approach can be applied to imitation of real-world videos (and if this is not the goal, please state this explicitly in the paper). Some evaluation of robustness due to variation in the demonstration videos (character width, color, etc) could have been helpful to assure the reader this approach could scale to real-world videos.

Second, only two demonstrations were showcased - 2D walking and 3D walking. It's hard to judge how this method (especially using RNNs to handle phase mismatch) would work for other motions.

Third, the evaluation to baselines is not adequate. Authors mention that GAIL does not work well, but hypothesize it may be due to not having a recurrent architecture. This really needs to be evaluated. A possibility is to set up a 2x2 matrix of tests between [state space, image space] condition and [recurrent, not recurrent] model. Would state space + not recurrent reduce to GAIL?

Fourth and most major to me is that looking at the videos the method doesn't actually work very well qualitatively, unless I'm misunderstanding the supplementary video. The tracking of 2D human does not match the style of the demonstration motion, and matches even less in 3D case. Even if other issues were to be addressed, this would still be a serious issue to me and I would encourage authors to investigate the reasons for this when attempting to improve their work.

Overall, I do not think the results as presented in the submission are up to the standards for an ICLR publication.

---

### Author Response · Authors · 2018-11-21
**Comments and baselines**

We would like to thank the reviewers for their great feedback.

The motivation for our work is two-fold. We understand that the dense pose or robot state space is a poor space to compare distances. We instead believe that a more appearance based distance function based on video will allow us to learn a better distance function and therefore policy. We focus on sequence-based distances as we believe that imitation often involves an inherent temporal structure that is not captured by current versions of GAIL. The extension that video-based methods may extend the imitation capabilities of robots even further is also a desired consequence.

Currently, we use rendered video data from the simulation. This gives us the possibility to continuously generate more video data. However, we are interested in extending this work to taking video input from other sources, for example, youtube or possibly the kinetics or NTURGB-D datasets. We leave this as exciting future work.

We have included additional motion tasks for imitation. These tasks include running backflipping and front flipping for the 3D biped. While the resulting policies are not of remarkable quality I would like to note that compared to prior methods published at SIGGRAPH we don’t provide motion phase information to the policy and our reward function is pure imitation and does not contain the many “hints” used in the DeepMimic/SVF/OpenAIGym environments for humanoid controllers.

Baselines:
We have processed additional comparisons to other baselines. These include a version of GAIL, non-recurrent vs recurrent examples and TCN examples. Most of these benchmarked on the 2d biped walking example. We are also working on an ablation study. We have also performed new comparisons to a multi-modal version of the recurrent method. Where we learn a distance function between the agents pose and the imitation video. This will also be included in the analysis.

Related work and details:
Thank you very much for the additional related papers.

We are currently editing the paper, this will all be included in an updated version of the paper in a few days. This updated version will also give many more details on the training process. In particular details on how the positive and negative examples are generated. In short we use a sequence-based version of a TCN like loss combine with class-by-class loss for the 3d humanoid where we have examples motions from other classes (running, backflipping and frontflipping).

We also thank the reviewers for their editing comments.

---

### Author Response · Authors · 2018-11-24
**New paper revision**

As requested, we have added numerous additional experiments which we outline below:

We have added two additional types of baseline comparisons. One comparing our method to GAIL and a VAE, and another comparing our method to a non-recurrent version that is similar to TCN. Please see figure 4a of the revised manuscript.

Based on these additional experiments we observe:
that training a VAE to learn an encoding to compute distances using an Euclidean norm is not effective.
Policies trained with GAIL often stand still or are jerky. Standing still is within the distribution of example imitation data.
Examining the new figure 4b:
We find that our method that takes advantage of temporal structure works well. Our method does not assume time alignment like many current TCN like methods and makes use of temporal structure that many current GAIL-based methods do not.
We also compare our method to the default manual reward function. We consistently find that both TCN and our method learn faster compared to learning with a manual reward function. We believe this is an indication that learned reward functions may provide a more dense reward landscape compared to the manual reward that can appear more sparse after the agent diverges from the desired behaviour.
We conduct an additional ablation analysis in figure 3c to compare the effects of particular methods used to assist in training the recurrent siamese network.
We find it very helpful to reduce Reference State Initialization (RSI). If more episodes start in the same state it increases the temporal alignment of training batches for the RNN.
We believe it is very important that the distance metric be most accurate for the earlier states in an episode so we use EESP (Early Episode Sequence Priority). Meaning we give the chances of cropping the window used for RNN training batches closer to the beginning of the episode. Also, we give higher probability to shorter windows. As the agent gets better the average length of episodes increases and so to will the average size of the cropped window.
Last, we tried pretraining the distance function. This leads to mixed results. Often pre training overfits the initial data collected leading to poor early RL training. However, in the long run pretraining does appear to improve over its non pretrained version.
We have also added experiments learning more tasks, including running and back/front flipping. We remark that:
The quality of the policy for running is reasonable (figure 6)
While the quality of the flipping is not as high as other methods it is important to remark that here we provide neither motion phase information to the policy nor any of the many “hints” that have been used to engineer other humanoid simulation RL techniques.
Still we are able to see some interesting behaviour that attempts to reproduce safer motions that match the motion timing -- in other words, our methods appear to learn in a ways closer to how real humans learn.

Thank you for your suggestions we have made numerous updates and added extensive imagery to illustrate these additions, as such the paper may take a long time to download. The paper is one page longer to accommodate the requested additional experimental work; however, we will work to compress the paper further should it be accepted.  Finally, we also provide a new video at the link below where we have added new results for the additional humanoid3d tasks:

Thanks for the various pointers to related work. We have added and discussed a number of them in the revised manuscript. Please note however that a number of these papers appeared after our paper was submitted.

New video of results can be found here: https://youtu.be/KGrDedTfclY

---

### Meta-Review · Area_Chair1 · 2018-12-14

**Confidence:** 4
**Recommendation:** Reject

**Metareview:**

This paper proposes an approach for imitation learning from video data. The problem is important and the contribution is timely. The reviewers brought up several concerns regarding the clarity of the paper and the lack of sufficient comparisons. The authors have improved the paper significantly, adding several new comparisons and improving the presentation. However, concerns still remain regarding the description of the method and the presentation of the results. Hence, the reviewers agree that the paper does not meet the bar for publication.